# Transcriptomic Analysis Reveals the Inability of Recombinant AAV8 to Activate Human Monocyte-Derived Dendritic Cells

**DOI:** 10.3390/ijms241310447

**Published:** 2023-06-21

**Authors:** Samer Masri, Laure Carré, Nicolas Jaulin, Céline Vandamme, Célia Couzinié, Aurélien Guy-Duché, Jean-Baptiste Dupont, Allwyn Pereira, Eric Charpentier, Laurent David, Gwladys Gernoux, Mickaël Guilbaud, Oumeya Adjali

**Affiliations:** 1Nantes Université, CHU Nantes, INSERM, TaRGeT—Translational Research in Gene Therapy, UMR 1089, F-44200 Nantes, France; 2Nantes Université, CHU Nantes, CNRS, INSERM, SFR Santé, UMS 3556, UMS016, F-44000 Nantes, France; 3Nantes Université, CHU Nantes, INSERM, Center for Research in Transplantation and Translational Immunology, UMR 1064, ITUN, F-44000 Nantes, France

**Keywords:** cell–virus interaction, AAV, innate immunity, gene therapy, human-monocyte-derived dendritic cells, transcriptomic analysis

## Abstract

Recombinant Adeno-Associated Virus (rAAV) is considered as one of the most successful and widely used viral vectors for in vivo gene therapy. However, host immune responses to the vector and/or the transgene product remain a major hurdle to successful AAV gene transfer. In contrast to antivector adaptive immunity, the initiation of the innate immunity towards rAAV is still poorly understood but is directly dependent on the interaction between the viral vector and innate immune cells. Here, we used a quantitative transcriptomic-based approach to determine the activation of inflammatory and anti-viral pathways after rAAV8-based infection of monocyte-derived dendritic cells (moDCs) obtained from 12 healthy human donors. We have shown that rAAV8 particles are efficiently internalized, but that this uptake does not induce any detectable transcriptomic change in moDCs in contrast to an adenoviral infection, which upregulates anti-viral pathways. These findings suggest an immunologically favorable profile for rAAV8 serotype with regard to in vitro activation of moDC model. Transcriptomic analysis of rAAV-infected innate immune cells is a powerful method to determine the ability of the viral vector to be seen by these sensor cells, which remains of great importance to better understand the immunogenicity of rAAV vectors and to design immune-stealth products.

## 1. Introduction

Recombinant adeno-associated virus (rAAV)-derived vectors are widely used for in vivo gene therapy. The long-term expression, efficacy and safety of gene transfer have enabled rAAV to move beyond preclinical research. Several rAAV products are currently being tested in clinical trials to target the liver, skeletal muscle or central nervous system [1,2,3] and six rAAV products have already been approved by European and/or American regulatory agencies for marketing (Glybera, Luxturna, Zolgensma, Roctavian, Hemgenix and Upstaza) [4,5,6,7,8,9]. Nevertheless, pre-existing immunity in humans following natural infection with a wild type AAV virus, as well as the immune responses against the viral vector itself after its in vivo delivery in gene transfer protocols, have risen a major concern during the translation of gene transfer protocols from animal models to humans [10,11,12]. In these clinical trials, the clearance of transduced cells was correlated with the detection of pre-existing rAAV capsid-directed T cells [10,11,12]. In another clinical trial aimed at correcting alpha antitrypsin (AAT) deficiency, capsid-specific gamma interferon (IFNg)-secreting cells were detected in rAAV1-injected patients using an ELISpot assay despite persisting transgene expression [13,14,15,16]. Although the injection routes, vector dosing and serotypes were not the same, these studies revealed that screening of peripheral cells using an IFNg ELISpot assay is not predictive of a deleterious cellular immune response. These different studies indicate that the impact of the cellular component of anti-AAV immunity on gene transfer remains not fully elucidated. When detected, AAV-capsid-specific T cells were evidenced using cellular proliferation or cytokine secretion assays without direct evidence of a cell cytotoxic activity. The modeling of the cellular response directed against the AAV vector capsid in animal models remains a challenge that needs to be addressed in order to better predict the response and better immunomodulate it. In addition, the diversity of gene transfer parameters in these studies using different administration routes, doses, serotypes in variable genetic background or animal species makes the investigation of rAAV cellular immunity difficult. Preclinical models will also have to reproduce the variability of the responses observed in humans [14,16,17].

One way to investigate AAV immunogenicity could be to focus on the response upstream of the cellular or humoral adaptative response. The initiation of an adaptive response in the host is subsequent to the recognition by the innate immune system of the viral vector as a pathogen or a non-self-antigen. This uptake step plays an important role in shaping the adaptive immune response towards rAAV antigens and may affect the success of gene therapy [18]. Understanding and modulating the initial steps of rAAV sensing by the innate system could be an effective way to control subsequent host immune responses against the viral capsid and or the transgene product.

The initiation of the innate response, in particular against viral-derived components, is mediated through pattern recognition receptors (PRRs) expressed on the membrane or in the cytosol of innate immune cells that recognize non-self-protein domains called pathogen-associated molecular patterns (PAMPs). Understanding the mechanisms of rAAV sensing requires the identification of the innate immune cells, together with the PRRs and PAMPs involved in vector recognition. Among the rAAV products, the viral DNA and capsid protein epitopes are potential PAMPs. The viral DNA can be sensed by different membrane and cytosolic sensors, such as the Toll Like Receptors 3 and 9 (TLR3, TLR9), cyclic GMP-AMP Synthase (cGAS), interferon-inducible protein 16 (IFI16), absent in melanoma 2 (AIM2), retinoic acid-inducible gene I (RIG-I). In contrast, capsid protein epitopes can be detected by other TLR that do not detect nucleic acids, such as TLR1, 2, 4, 5 and 6 [19]. Among the cells involved in the innate response, plasmacytoid dendritic cells (pDCs) with the help of conventional DCs (cDCs) have been identified as key sensor cells for the recognition of rAAV2 in in vitro and in vivo mice models of gene transfer [20,21]. This recognition involves the PRR TLR9 and its adapter myeloid differentiation primary response 88 (MyD88), which detect vector DNA and trigger the type I interferon pathway. By another approach, Hösel et al. showed that AAV serotype 8 can be recognized in vitro by primary human liver cells [22]. In this case, AAV8 capsid proteins were detected by the TLR2 of non-parenchymal liver cells (e.g., Kupffer and sinusoidal endothelial cells) but not by hepatocytes. In contrast, Shao et al. showed that rAAV2 double-stranded-derived transcripts could also be detected by primary human hepatocytes in vitro and in vivo in a humanized liver mouse model [23]. AAV genome detection was shown to be mediated by the cytosolic sensor melanoma differentiation-associated protein 5 (MDA5) previously identified to recognize double-stranded RNA.

Altogether, these different yet few studies highlight the complexity of rAAV sensing. The recognition of vector different PAMPs seems to use various PRR and pathways which may depend on several factors, such as the AAV serotype, the genome conformation (single-stranded versus self-complementary), the vector dose, the innate cells involved in the target tissue, and therefore the administration route.

Here, we aimed at studying the mechanisms of rAAV sensing in human DCs using monocyte-derived dendritic cells (moDCs). We have chosen the AAV8 serotype because it is already in clinics and shows great promise for the treatment of various diseases (hemophilia B, myopathies) [24,25] and because the recognition and immunogenicity of rAAV8 products are not yet clearly established. A number of preclinical studies in rodents as well as large animal models have reported a poor immunogenicity for rAAV products of serotype 8 [26,27,28,29,30]. Our results indicate that (i) rAAV8 is efficiently internalized into human moDCs but results in poor transduction efficacy, (ii) rAAV8 does not elicit the upregulation of activation surface markers on moDCs, and (iii) rAAV8 does not induce significant changes in the transcriptome of moDCs. In summary, rAAV8 seems to be unable to elicit any inflammatory or anti-viral response in human moDCs, which may provide an immune stealth status to this particular serotype.

## 2. Results and Discussion

### 2.1. AAV8 Is Efficiently Internalized into Human moDCs

To study the impact of rAAV8 on the human innate immune system, moDCs were generated in vitro from human monocytes cultured in the presence of interleukin 4 (IL-4) and granulocyte macrophage colony-stimulating factor (GM-CSF). To ensure proper differentiation (CD14^−^/HLA-DR^+^/CD209^+^), the moDC phenotype was systematically assessed by flow cytometry before incubation with rAAV. Phenotyping results are presented in Appendix A and show that 98.8% of the cells were CD14^−^ while more than 98.4% of the CD14^−^HLA-DR^+^ cells expressed the DC sign marker (CD 209) specific to dendritic cells.

After phenotype validation, human moDCs from three donors were incubated with single-stranded (ss) rAAV8 CMV GFP, harvested at 24 h post-infection and stained with the membrane marker wheat germ agglutinin (WGA) and AAV8-capsid-specific antibody (ADK8), which binds specifically on the protrusion facing the 3-fold axis of rAAV8 capsids to visualize the vector localization [31,32]. This aims at visualizing rAAV8 viral particles and determine their localization and potential uptake by moDCs. Confocal microscopy z-stack image acquisition showed that rAAV8 is internalized in a large number of cells either in the nucleus or in the cytoplasm (Figure 1a and Appendix A). To confirm this observation, the internalization rate of rAAV was quantified for the three donors by computer image processing (Table 1). This showed that 53 to 98.7% of moDCs internalized the vector.

Then, human moDCs were incubated with ss rAAV8 CMV GFP (cytomegalovirus promoter and green fluorescent protein transgene) or adenovirus CMV GFP (Ad), or were non-treated (NT) to assess vector transduction efficiency. Cells were harvested at 48 h post-infection and reporter transgene (GFP) expression was examined by fluorescence microscopy. No GFP expression was detected in cells incubated with rAAV8 compared to recombinant adenovirus vector (Figure 1b). To confirm the lack of transgene expression, levels of GFP were quantified by flow cytometry on moDCs from seven donors (Figure 1c and Appendix A). When transduced by Ad vectors, 81% ± 12 SD of human moDCs expressed the GFP protein while AAV CMV GFP transduction led to only 1% ± 0.1 SD detectable fluorescent cells, similar to the NT control condition (*p*-value between Ad and rAAV vectors = 0.0006) (Figure 1c).

To further confirm the inefficiency of rAAV8 to transduce MoDCs, total RNA was extracted from moDCs of four donors, and transgene transcripts were quantified by RT-qPCR at 48 h post-infection. Results are expressed as relative quantity (RQ) of transgene transcripts compared to the reference SDHA gene (succinate dehydrogenous complex, subunit A). The RQs comprised between 45 and 396 Ad-infected moDCs. For rAAV8-transduced moDCs, the RQ values were found to be very low (between 0.02 and 0.09), although still above the threshold of qPCR detection (determined at 0.00461) (Figure 1d).

The detection of internalized rAAV8 capsids in moDCs but only low levels of transgene mRNA indicate that rAAV8 particles are capable of entering moDCs but that their intracellular processing to enable transgene expression is somewhat inefficient. The level of RNA transgene transcripts was monitored for 72 h after moDC incubation with rAAV8 (Appendix A). The results indicate that although low, the peak mRNA level is reached as early as 24 h post-infection with no increase thereafter despite the presence of uncoated AAV8 particles in the cells. One potential mechanism of rAAV8 restriction could be an inefficient uncoating of a large proportion of intracellular rAAV8 viral genomes, a mechanism that should occur early within the first 24 h after vector entry [33].

To further investigate this hypothesis, moDCs from three human donors were incubated with GFP-encoding rAAV8 or adenovirus vectors for 24, 48 and 72 h. Unfortunately, due to a limited number of cells, conditions moDC + Ad at 48 and 72 h could only be performed for one out of the three donors. For each condition, the cell pellet was divided into two fractions, and total DNA was extracted with or without proteinase K (PK). The treatment with PK allows for the extraction of the total DNA. In contrast, in the absence of proteinase K, only DNA that is already capsid-uncoated is extracted. Vector DNA was measured by qPCR and normalized with SDHA gene. Results were expressed as the delta of vector genomes in the presence of PK minus in the absence of PK (Figure 2). The difference in the amounts of viral genomes between the (+PK) and (−PK) conditions is almost zero when moDCs are incubated in the presence of the adenovirus. This indicates that most adenoviral genomes are extracted and available for quantification even in the absence of proteinase K. In contrast, when moDCs are incubated with rAAV8, the difference observed in quantified viral genomes in the presence of PK and those in the absence of PK indicates that a fraction of the rAAV viral genomes is not quantified when DNA extraction was performed in the absence of PK. Moreover, this difference between (+PK) and (−PK) conditions is observed for at least 72 h post-infection and reinforces the hypothesis that part of the rAAV capsids remains intact in dendritic cells. However, despite this difference, it is important to note that in the absence of PK, about 50% of the DNA signal is measured (Appendix A) which indicates that the restriction is not only limited to this hypothesis.

In the literature, few data are available on rAAV8 and its ability to transduce dendritic cells and, in particular, moDCs. Most studies have used rAAV1 and rAAV2 and have shown that they can efficiently transduce human or mouse DCs in vitro [12,34,35,36]. Regarding AAV8, there are studies, showing a lack of immunogenicity of rAAV8 in animal models [26,27,28,29,30], which suggest a poor transduction efficacy of antigen-presenting cells by this particular serotype. This remains controversial. Indeed, two previous studies showed an in vitro transduction of mouse DCs or human moDCs with poor efficiency compared to rAAV1 or rAAV2 [37,38]. Other studies have reported the ability of rAAV8 to transduce DCs in vitro [28] and in vivo, particularly spleen follicular DCs in mice and nonhuman primates [39,40]. In our study, the low detection of transgene-derived proteins and/or transcripts despite the high internalization rates of rAAV8-intact particles in moDCs suggests a postinternalization restriction mechanism as already described by Rossi et al. for the AAV2 serotype [38]. Aside from the hypothesis of an uncoating defect as discussed above, this restriction can be related to a failure in any other step of rAAV8 intracellular processing, such as trafficking impairment, defect in endosomal escape, partial nuclear import, accumulation of the vector in the nucleolus or a defect of the second DNA strand synthesis (reviewed by Berry and Asokan [41]). Further investigation is needed to better characterize the mechanisms of intracellular restriction of the rAAV8 vector in moDCs.

As we have shown here that rAAV8 can be internalized in moDCs, we next wanted to assess its ability to activate DCs, which is a prerequisite to trigger an immune response.

### 2.2. Recombinant AAV8 Fails to Activate Human moDC Inflammatory Pathways In Vitro

Our data show the internalization of AAV in a large proportion of moDCs. To evaluate the impact of such vector entry on the activation of these antigen-presenting cells, we therefore analyzed the expression of moDC costimulatory molecules CD80 and CD86 after rAAV8 CMV GFP incubation. In parallel, to validate the ability of the moDC model to upregulate costimulatory molecules, moDCs were incubated with a potent TLR4/7/8 activator, the lipopolysaccharide/resiquimod cocktail (LPS/R848) or with a more physiological immunogenic viral stimulation, such as the recombinant GFP-expressing adenovirus (Ad CMV GFP). MoDCs collected from seven donors were harvested 48 h post-incubation and the proportions of CD80+- and CD86+-activated cells were measured by flow cytometry (Figure 3, Appendix A). In the presence of LPS/R848, we observed an upregulation of CD80 and CD86 markers in 94.5 and 68.2% of cells, respectively, compared to the non-treated condition (NT). Adenoviral and rAAV conditions were found statistically different (*p*-values of 0.0012 and 0.0006 for CD80 and CD86 markers, respectively). In the presence of the adenoviral vector, we observed an upregulation of CD80 and CD86 markers in 9.03% and 27.9% of moDCs, respectively, compared to the NT control. Conversely, no difference in CD80/CD86 marker expression was observed in cells incubated with rAAV8 compared to non-infected cells.

Our data suggest that rAAV8 does not activate primary human moDCs, which is consistent with the results obtained by Mays et al. in mice [42]. In this study, the authors showed that rAAV8 delivery is not able to induce the activation of DCs collected from draining lymph nodes, in contrast to mice injected with the rAAV Rh32-33 serotype, for which DCs expressed high levels of CD80/86. This particular serotype was reported to be immunogenic in contrast to AAV8 by the same team [29], and it has been recently considered as a candidate for the development of rAAV-based vaccines [43]. Nonetheless, not all studies have shown a lack of DC activation by rAAV8. In dogs, the rAAV8-mediated transduction of bone-marrow-derived DCs (BMDCs) was shown to induce CD80 and MyD88 upregulation after 4 h of incubation [28]. In this case, marker upregulation was shown to be far less than that caused by infection with rAAV2. In conclusion, the lack or weak upregulation of activation markers on the surface of dendritic cells may impair the initiation of adaptive immune response and further confirms the reduced immunogenicity of the AAV8 serotype.

As rAAV8 does not upregulate the standard activation markers after transducing moDCs, we asked whether other markers and associated alternative pathways of antigen uptake and cell activation could be upregulated by rAAV8 infection. For this purpose, we analyzed the moDC transcriptome of 12 healthy human donors after incubation with rAAV8 CMV GFP, adenovirus CMV GFP, LPS/R848, or medium (NT). The total RNA was extracted and sequenced by 3′ sequencing RNA profiling (3′SRP), a quantitative RNA sequencing method based on unique molecular identifiers (UMIs), to determine the absolute copy number of each messenger RNA [44].

A principal component analysis (PCA) was generated by bioinformatic processing to group samples with a similar gene expression profile (Figure 4a). The PCA revealed three distinct groups of samples: moDCs incubated in presence of LPS/R848, moDCs incubated in presence of the adenoviral vector and a third group with the NT controls as well as the rAAV-transduced moDCs. The transcriptome profile of rAAV-transduced moDCs seems not to be significatively different from non-transduced cells. To confirm this, the RNA sequencing data were also projected on a heat map to highlight significantly upregulated (red) and downregulated (blue) genes and also to group samples with similar clusters (Figure 4b). When moDCs were incubated in the presence of TLR4, 7 and 8 activators (LPS/R848), we observed a strong upregulation of one gene cluster (in red) and a strong downregulation of one gene cluster (in blue). Similarly, in the presence of the adenoviral vector, the same profile was found but to a lesser extent, as only some small clusters of up- and downregulated genes were highlighted. The LPS/R848 samples, on the one hand, and the Ad-treated samples, on the other hand, are grouped into two distinctive groups with similar activation profiles as indicated by the dendrograms in the figure. Finally, in the presence of rAAV, we did not evidence a difference in gene expression between the rAAV8-transduced and NT control groups, making the samples unidentifiable by their gene expression profile. In our model, despite its entry, rAAV8 does not seem to induce any modification of the moDC transcriptome. To validate the relevance of our analysis and cell model used, we looked more precisely at the moDC transcriptomic profile in the presence of the adenoviral vector (Appendix A). The resulting volcano plot identified upregulated genes and downregulated genes (Appendix A). Interestingly, among the 20 highly upregulated genes identified on the volcano plot (Appendix A), 12 correspond to interferon-induced proteins (interferon-alpha-inducible protein (27-IFI27), interferon-induced protein with tetratricopeptide repeats (2-IFIT2), interferon-induced protein with tetratricopeptide repeats (3-IFIT3), interferon-induced transmembrane protein (1-IFITM1), interferon-stimulated gene (15-ISG15), 2’-5’-oligoadenylate synthetase-like (OASL) protein, interferon-induced protein with tetratricopeptide repeats (1-IFIT1), ubiquitin-specific peptidase (18-USP18), interferon-induced GTP-binding protein (MX1), radical S-adenosyl methionine domain-containing (2-RSAD2), interferon-alpha-inducible protein (IFI6), interferon-induced protein 44-like (IFI44L), interferon regulatory factor (7-IRF7)). Gene ontology (GO) analysis was then used to determine the molecular functional significance of the upregulated genes (Appendix A). Among the 9 most relevant annotations, the GO analysis revealed three significantly enriched terms: (i) the activation of the type I interferon pathway (GO:0060337), (ii) the negative regulation of viral genome replication (GO:0045071) and (iii) the triggering of the antiviral immune innate response (GO:0140374). These findings indicate that the moDCs generated in vitro are functional and are able to recognize and trigger an innate inflammatory response against virus-derived components. Importantly, a similar analysis on cells incubated with rAAV8 did not result in any differentially expressed genes, which further confirms that the AAV vector does not lead to any detectable changes in moDC transcriptome.

In more details, our transcriptomic results showed that in human moDCs, rAAV8 does not activate any of the described TLR-dependent or TLR-independent pathways. In contrast, several studies using rAAV1, 2 and 9 reported an activation of dendritic cells by AAV vectors through the TLR9/MyD88 pathway using in vitro and in vivo mouse models [20,21]. Additionally, Hösel et al. showed that TLR2 could be involved in the detection of the vector capsids 2 and 8 in human nonparenchymal liver cells [22]. Several factors may explain the discrepancies between the present study and the literature. We used human primary cells, while most studies have been performed in mouse models. Moreover, the presence of TLR9 on monocyte-derived human DCs, as used in this study, is not yet clearly established as some studies indicate that human moDCs express TLR9 [45], while others do not [46]. In the present study, we did not detect TLR9 gene expression in our cells, which may explain the lack of moDC reactivity to rAAV8. The fact that our cells properly reacted to adenoviral infection used as an “activating control virus” indicates nonetheless that the MoDCs used in the study are able to respond to a viral signal as annotated by ontology (Appendix A).

Many other activation pathways alternative to TLR9/MyD88 were described for viral sensing and could be involved in rAAV detection after its cellular uptake, such as TLR2, MDA5, stimulator of interferon genes (STING), cGAS and IFI16 [19,22,23,47,48]. In our experimental model, none of these pathways were upregulated or changed despite the internalization of rAAV8 particles. One possibility is that the upregulation of such pathways with rAAV is early and only transient. Indeed, several studies have described a fast rAAV-induced activation of dendritic cells as early as 2 to 6 h post-incubation [49] or at 18 h [20]. In this latter study, Zhu et al. [20] have shown that IFNa secretion from pDCs in response to AAV2 remains relatively stable until at least 48 h. In our transcriptome study, no significant difference was observed between the 24H and 48H incubation times (Figure 4b). The assessment of rAAV8 cellular activation at earlier time points could therefore be of interest.

This study was performed with AAV serotype 8 because it is clinically widely used for various systemic applications, such as hemophilia B, myopathies and metabolic deficits. To date, few data are available on the interaction of rAAV8 serotype with the innate immune system in contrast to the historical AAV2 serotype. Here, the lack of detectable moDC activation by rAAV8 further confirms the poor immunogenicity of this specific serotype as reported in previous preclinical studies in large animal models after vector systemic delivery [26,30]. In a human clinical trial, Nathwani and colleagues have shown that rAAV8 can induce humoral and, to a lesser extent, cellular immunity directed against the capsid [24] that likely results from vector detection by other antigen-presenting cells, such as macrophages. Despite this, transgene expression persisted over time in patients, up to 30 weeks. The efficacy and intensity of the rAAV vector immunity may vary between different serotypes as emphasized by hemophilia trials [8,50]. In the case of the AAV8 serotype, host-adaptive immunity could not be effective enough to eliminate the vector. The AAV8 serotype may have developed an escape mechanism from the immune system, similar to some other viruses, such as hepatitis B virus (HBV), herpes simplex virus-1 (HSV-1) or Kaposi’s sarcoma-associated herpesvirus (KHSV), which are able to escape their detection [51]. On the other hand, Mays et al. provided evidence of a state of tolerance towards rAAV8. Indeed, rAAV8 delivery to the muscle of C57BL/6 mice resulted in poor APC transduction and the lack of upregulation of CD80/86 in transduced cells. In addition, the absence of upregulation of major histocompatibility complex (MHC) I and II in these antigen-presenting cells limited CD8^+^ T cell priming and prevented proper antigen display on target cells [42]. These results are in agreement with our findings regarding dendritic cell activation and priming. However, our data need to be confirmed in models recapitulating the complexity of the response. The activation of DCs by rAAV may indeed require their cooperation with additional immune actors, such as other cell types, antibodies or the complement system response, as shown in [52,53,54,55].

In conclusion, in this study, we developed a methodology to characterize the impact of rAAV on primary human moDCs using RNAseq-based technology. Our results showed that part of the rAAV8 particles is able to enter moDCs and reach the nucleus to start a transcription process, while the other part remains trapped in moDCs without being uncoated. Despite this, AAV8 does not appear to trigger immune activation in moDCs as we have shown using an integrative transcriptomic-based method. Future studies will need to address the mechanisms of rAAV8 immune recognition in DCs, using additional models to recapitulate the complexity of both the immune system and the viral vector biology. That being said, transcriptomic analysis of rAAV-infected innate immune cells is a powerful method to determine the ability of the viral vector to be seen by these sensor cells, which remains of great importance to better understand the immunogenicity of rAAV vectors and design immune-stealth products.

## 3. Materials and Methods

### 3.1. Generation of Monocyte-Derived Dendritic Cells (moDCs)

Monocytes of 12 human voluntary and healthy donors were provided by IPSCDTC Core facility of Nantes. Fresh or frozen monocytes were differentiated in moDCs for 7 days at 37 °C in RPMI medium supplemented with 10% fetal bovine serum (GE HealthCare, Chicago, IL, USA), 1% penicillin/streptomycin (ThermoFisher Scientific, Waltham, MA, USA), 100 ng/mL of rGM-CSF (15 µL), 20ng/mL of rIL-4 (3 µL) and 20 mL/mL) of human serum albumin (1.5 mL).

### 3.2. Vector Production

Recombinant AAV batches were produced by the Center for Production of Vectors of Nantes (TaRGeT, Nantes, France) in human embryonic kidney 293 cells through cotransfection with (i) the vector plasmid containing the GFP sequence (eGFP variant), the cytomegalovirus (CMV) promoter and the SV40 polyadenylation signal, flanked by two AAV2-ITRs, and (ii) the pDP8 plasmid containing viral sequences required for replication and encapsulation [56]. After supernatant precipitation with polyethylene glycol, vectors were purified by a cesium chloride gradient according to Ayuso et al. [56] and concentrated using the Apollo system (BD Biosciences, Franklin Lakes, NJ, USA). Quantitative PCR was performed on ITR sequences as previously described by D’Costa et al. [57], and titers were expressed in viral genomes/milliliter.

### 3.3. Staining of rAAV8 Particles by Immunohistochemistry

Monocyte-derived dendritic cells were infected by rAAV8 GFP at a multiplicity of infection (MOI) of 10^6^ vector genome per cell or recombinant adenovirus expressing the eGFP under the control of the CMV promoter at MOI = 500 infectious particles per cells. For adenovirus, the MOI was determined as the lowest, for which a maximum GFP signal is detected, whereas for rAAV8, MOI = 10^6^ was used as it is the maximum possible MOI per cell in this assay. Sixteen hours later, moDCs were stained firstly using WGA (for membrane labelling), then fixed by paraformaldehyde 4% (ThermoFischer Scientific, Waltham, MA, USA) for 10 min and washed 3 times with PBS. Non-specific binding sites were saturated with PBS 10% goat serum (Merck, Darmstadt, Germany) for 20 min, and then incubated with ADK8 anti-AAV8 capsid antibody for 1 h at room temperature (RT). ADK8 is a hybridoma supernatant that was kindly provided by Jürgen Kleinschmidt lab (Heidelberg, Germany). After a washing step, goat anti-mouse Alexa-555 secondary antibody (ThermoFischer Scientific, Waltham, MA, USA) was incubated for 1 h at RT followed by nucleus staining with DAPI (Merck). Stained cells were dropped on glass slides and air-dried at RT for 20 min. Finally, one drop of Prolongold (ThermoFischer Scientific, Waltham, MA, USA) was added on cells and slides were incubated at RT in the dark for 48 h until the complete drying of Prolongold. Cell images were acquired using an Apotome confocal A1 microscope (Nikon, Tokyo, Japan) at the MicroPICell platform (Nantes, France).

### 3.4. Image Analysis

The analysis was performed using Fiji software (v 2.12) [58]. Presence of AAV inside nucleus and cytoplasmic compartments was monitored using the particle analysis function after automatic thresholding.

### 3.5. Quantification of Transgene mRNA by RT-qPCR

Monocyte-derived Dendritic Cells were infected by rAAV8 GFP or by recombinant Adenovirus GFP as previously described. As controls, moDC were incubated with lipopolysaccharide (LPS, Merck, Rahway, NJ, USA)/Resiquimod (R848, Miltenyi Biotec, Bergisch Gladbach, Germany)—TLR4/TLR7/TLR8 activation cocktail (0.1 µg/mL of LPS and 1 µg/mL of R848) or in medium for non-treated control (NT). Forty-eight hours later, cells were harvested and frozen in liquid nitrogen. Total RNA was obtained from lysis of cell using TRIzol^®^ (ThermoFisher Scientific, Waltham, MA, USA) and chloroform-based (Merck, Darmstadt, Germany) extraction. Briefly, 1 mL of TRIzol was added right after thawing to the frozen cell pellet to avoid RNA degradation, then after homogenization, 200 µL of chloroform was added. After a centrifugation step at 12,000× *g* at +4 °C, the upper aqueous phase was collected and mixed with glycogen and isopropanol. After precipitation, washing and Nanodrop quantification (ThermoFisher Scientific, Waltham, MA, USA), total RNA was then treated with RNAse-free DNAse I from TURBO DNA-freetM Kit (ThermoFischer Scientific, Waltham, MA, USA) to eliminate DNA contamination. RNA was then reverse-transcribed in cDNA using M-MLV RTase kit according to manufacturer recommendations (ThermoFisher Scientific, Waltham, MA, USA). Vector and endogenous transcripts were measured through quantification of cDNA using a StepOne plus thermocycler (Applied Biosystems, Waltham, MA, USA). The primer and TaqMan Probe used for GFP cDNA quantification were: forward primer 5′-ACTACAACAFCCACAACGTCTATATCA-3′, reverse primer 5′-GGCGGATCTTGAAGTTCACC-3′ and probe 5′-(6FAM)-CCGACAAGCAGAAGAACGGCATCA-(TAMRA)-3′. The GFP qPCR was performed with the following program: initial denaturation 20 s at 95 °C followed by 45 cycles of 1 s at 90 °C and 20 s at 60 °C.

The reverse-transcribed mRNA measurement was normalized by quantifying the endogenous SDHA reverse-transcribed mRNA using the following primers to target the SDHA sequence: forward primer 5′-CTTGCGAGCTGCATTTG-3′ and reverse 5′-CCCAGAGCAGCATTGAT-3′. The SDHA qPCR was performed using SYBRgreen technology (Takara) and used the following program: initial denaturation for 20 s at 95 °C, followed by 40 cycles of 30 s at 90 °C, 30 s at 60 °C, and 1 melt stage of 95 °C for 1 s, 62 °C for 1 min and 95 °C for 1 sec. The Ct results obtained for the transgene transcripts were normalized with SDHA Ct values to obtain a Relative Quantity: RQ = 2^−ΔCt^ where ΔCt = Ct_transgene − Ct_endogenous.

### 3.6. Quantification of Vector Genome Extracted with or without Proteinase-Capsid Degradation

Monocyte-derived dendritic cells were infected with rAAV8 GFP or by recombinant adenovirus GFP as previously described. 24, 48 and 72 h later, cells were harvested and split in two fractions. Total genomic DNA (gDNA) was isolated from cells using the Gentra Puregen kit (Qiagen, Hilden, Germany) according to the manufacturer’s recommendations, with the exception of proteinase K, which was used on only one of the two fractions collected for each sample, fractions (−PK) and (+PK). The amount of genomes was estimated by qPCR using 50 ng of gDNA as an input and by targeting the GFP sequence and the SDHA gene for normalization as described for transgene mRNA quantification. Results were expressed in vector genomes per diploid genomes (vg/dg).

### 3.7. Cell Phenotyping by Flow Cytometry

A phenotypic analysis of human moDCs was performed for each donor after the differentiation of monocytes. Cells were stained using Zombie NIR viability dye, anti-CD14 (M5E2, BD Biosciences, Franklin Lakes, NJ, USA, RRID:AB_2687593), anti-CD209 (DCN46, BD Biosciences, Franklin Lakes, NJ, USA, RRID:AB_394123), and anti-HLA-DR (L243, BD Biosciences, Franklin Lakes, NJ, USA, RRID:AB_2738559). Cells were acquired using LSR II cytometers (BD Biosciences, Franklin Lakes, NJ, USA), and results were analyzed using FlowJo softwares (v10, Treestar, Woodburn, OR, USA). The activation state of moDC after their infection with rAAV was analyzed by flow cytometry using the following antibodies: anti-CD80 (L307.4, BD Biosciences, RRID:AB_10562564), anti CD86 (2331, BD Biosciences, Franklin Lakes, NJ, USA, RRID:AB_11153866).

### 3.8. 3′Sequencing RNA Profiling (3′SRP)

3’seq-RNA profiling protocol was performed according to [44]. The libraries were prepared from 10 ng of total RNA in 4 µL. The mRNA poly(A) tails were then tagged with universal adapters, well-specific barcodes and unique molecular identifiers (UMIs) during template-switching reverse transcription. Barcoded cDNAs from multiple samples were then pooled, amplified and tagmented using a transposon-fragmentation approach, which enriches the 3’ends of cDNA: 200 ng of full-length cDNAs were used as input to the Nextera™ DNA Flex Library Prep kit (ref #20018704, Illumina, San Diego, CA, USA) and Nextera™ DNA CD Indexes (24 indexes, 24 samples) (ref #20018707, Illumina) according to the manufacturer’s protocol (Nextera DNA Flex Library Document, ref #1000000025416 v04, Illumina).

The size of library was controlled on 2200 Tape Station System (Agilent Technologies, Santa Clara, CA, USA). A library of 350–800 bp length was run on a NovaSeq 6000 using NovaSeq 6000 SP Reagent Kit 100 cycles (ref #20027464, Illumina) with 17*-8-105* cycles read. * shows that the addition of an additional cycle to the reads when sequencing on the NovaSeq is recommended by Illumina.

### 3.9. Bioinformatics Protocol

Raw fastq pairs match the following criteria: the 16 bases of the first read correspond to 6 bases for a designed sample-specific barcode and 10 bases for a unique molecular identifier (UMI). The second read (104 bases) corresponds to the captured poly(A) RNA sequence. The dataset generated for this study is available on Gene Expression Omnibus (GEO) from the National Center for Biotechnology Information (GEO accession number: GSE228649). Bioinformatic analysis was performed using a snakemake [59] pipeline (https://bio.tools/3SRP, accessed on 6 August 2019). Sample demultiplexing was performed with a python script. Raw paired-end fastq files were transformed into a single-end fastq file for each sample. Alignment on refseq reference transcriptome, available from the UCSC download site, was performed using bwa. Aligned reads were parsed, and UMIs were counted for each gene in each sample to create an expression matrix containing the absolute abundance of mRNAs in all samples. Reads aligned on multiple genes or containing more than 3 mismatches with the reference were discarded. The expression matrix was normalized, and differentially expressed genes (DEG) were searched using the R package limma and edgeR [60,61]. Gene ontology (GO) was used to provide insights into finding a biological hypothesis.

## Figures and Tables

**Figure 1 ijms-24-10447-f001:**
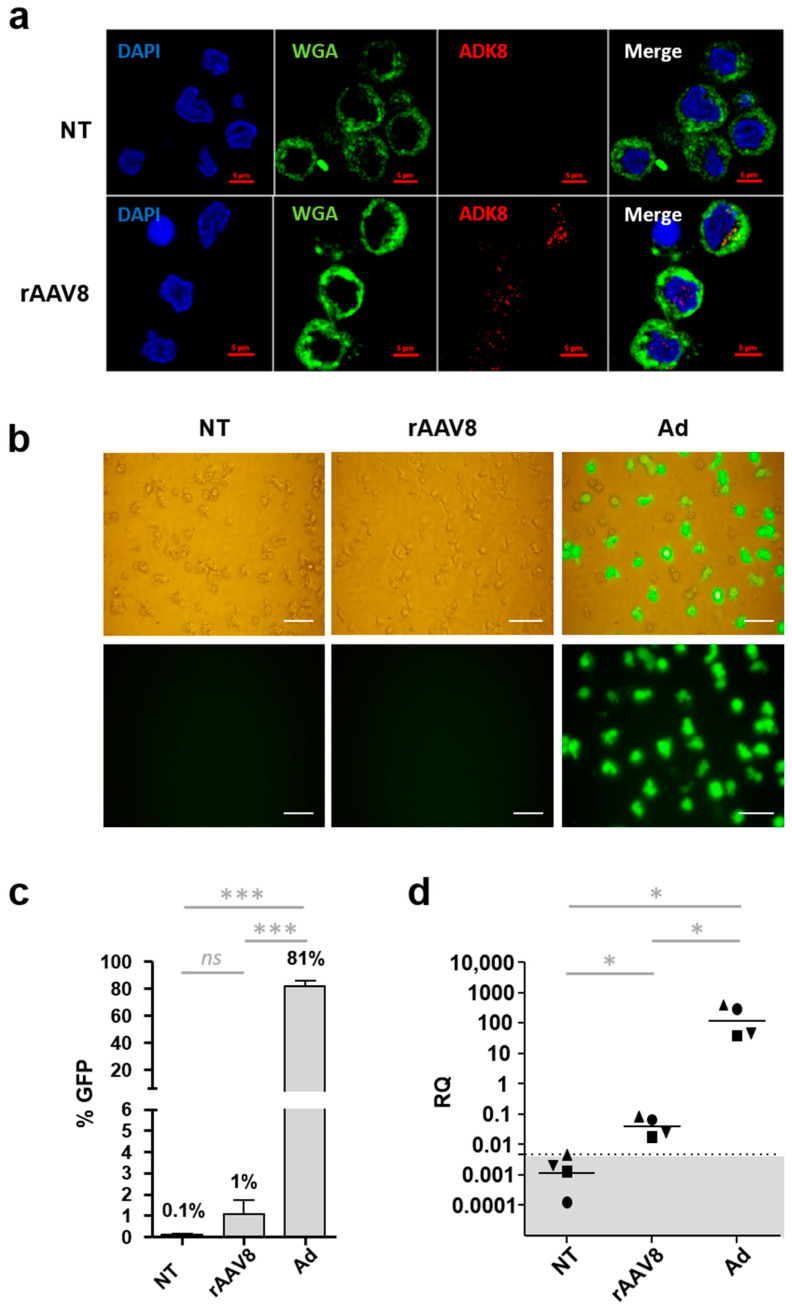
rAAV8 entry and transduction of human moDCs. (**a**) Intracellular detection of rAAV8. Monocyte-derived DCs (moDCs) were incubated with single-stranded recombinant AAV vectors (rAAV8) encoding for GFP at a MOI of 10^6^ for 24  h, or non-treated (NT). Cells were harvested, fixed, permeabilized, and incubated with antibodies directed against the AAV capsid (ADK8) and membrane dye WGA, respectively. The nucleus was stained with 4′,6-diamidino-2-phenylindole (DAPI). Cells were imaged by confocal microscopy (*n* = 3). Scale bar: 5 µm (**b**) Detection of GFP-expressing MoDCs. Cells were collected at 48 h post-infection after incubation with rAAV8 CMV GFP vector (rAAV8), adenoviral vector CMV GFP (Ad) or NT. GFP expression was observed by direct fluorescence microscopy. Scale bar: 50 µm (**c**) Quantification of GFP-expressing moDCs. MoDCs were transduced by rAAV8, Ad or NT, and expression of GFP was measured by flow cytometry. Data are presented as the mean percentage of positive cells (*n* = 7). (**d**) Quantification of transgene transcripts. Monocyte-derived DCs were harvested 48 h post-transduction, and transgene transcripts (GFP) were detected by qRT-PCR. Results are expressed as the mean of relative quantity (RQ) (*n* = 4). The dotted line represents the qPCR limit of quantification determined at 0.00461. Non-parametric Mann–Whitney statistical test were performed between different groups and significant differences were reported as *** and * for *p*-values < 0.001 and <0.05, respectively. Non-significant differences were noted as ns.

**Figure 2 ijms-24-10447-f002:**
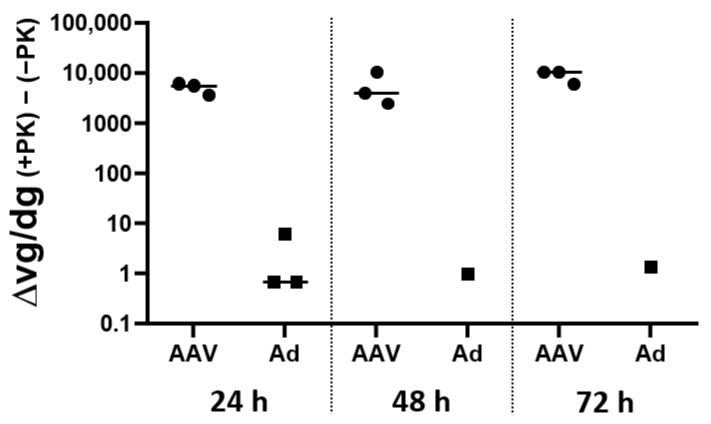
A proportion of rAAV genomes remains uncoated in moDCs. Monocyte-derived dendritic cells from three human donors were incubated with GFP-encoding rAAV8 or adenovirus vectors for 24, 48 and 72 h. Due to a limited number of cells, conditions moDC + Ad at 48 h and 72 h could only be performed for one out of three donors. Cell pellets were split in two fractions, and total DNA was extracted in presence or in absence of proteinase K (PK) and vector genomes were quantified by qPCR, normalized with SDHA gene and expressed in vector genome per diploid genome (vg/dg). The delta of genome quantification in presence of PK minus in absence of PK (Δvg/dg (+PK) − (−PK)) was calculated in order to highlight capsid-trapped genomes unreleased by PK.

**Figure 3 ijms-24-10447-f003:**
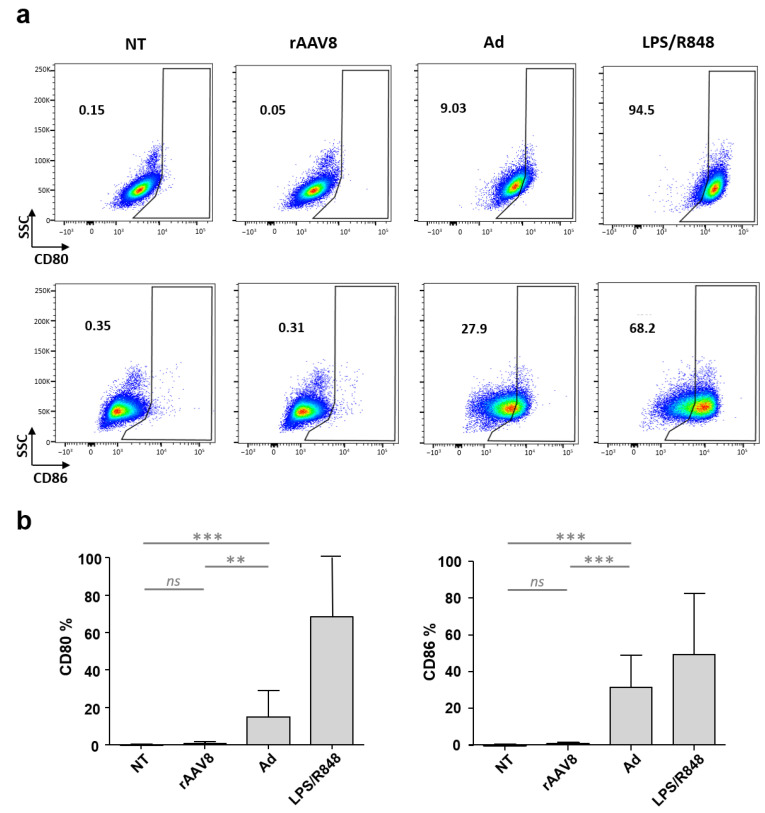
rAAV8 does not upregulate moDC activation markers. Analysis of human moDCs incubated with rAAV8, Ad, LPS/R848 or NT. Cells were harvested at 48 h and the expression of costimulatory molecules (CD80/CD86) was quantified by flow cytometry. (**a**) Dot plots of costimulatory molecules of a representative donor of moDCs. Blue and green correspond to areas of lower cell density, red and orange are areas of high cell density, and yellow is mid-range (**b**) Mean percentage of each costimulatory molecule in human moDCs (*n* = 7 donors). Non-parametric Mann–Whitney statistical test were performed between different groups, and significant differences were reported as *** and ** for *p*-values < 0.001 and 0.01, respectively. Non-significant differences were noted as ns. SSC: side scatter.

**Figure 4 ijms-24-10447-f004:**
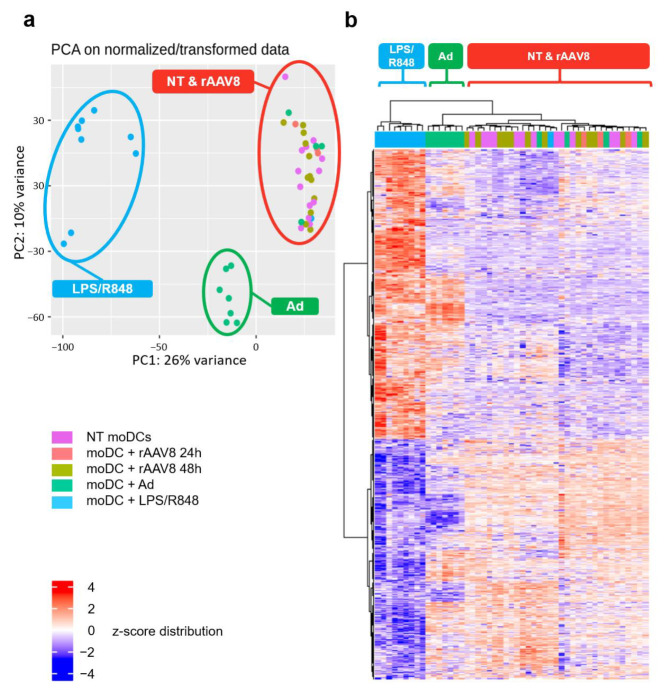
rAAV8 has no transcriptomic impact on human moDC gene expression 48 h post incubation. Human moDCs from 12 healthy donors were infected with rAAV8 GFP (rAAV8), recombinant adenovirus GFP (Ad), LPS/R848 or non-treated (NT). Certain cells were harvested at 24 and 48 h post-infection (rAAV8) and others at only 48 h post-infection (Ad, LPS/R848 and NT). Cellular RNA was extracted and dosed, and its RNA integrity number (RIN) was measured. (**a**) Principal component analysis (PCA) shows good agreement with sample correlation: negative controls NT and rAAV8 are distinctly separated from Ad- and LPS/R848-activated control conditions. (**b**) Heatmap showing scaled gene expression (log TPM values) of discriminative gene sets defining each moDC condition is represented. Color scheme is based on *z*-score distribution, from −4 (Blue) to 4 (Red) expression distributions of genes across moDC conditions on the *y*-axis.

**Table 1 ijms-24-10447-t001:** Percentage of cells with internalized rAAV8.

Donor	Number of Analyzed Cells	Number of Cells with Internalized rAAV8	% of Cell with Internalized rAAV8
A	1796	1774	98.7%
B	221	183	82.8%
C	345	183	53%

## Data Availability

The dataset generated for this study is available on Gene Expression Omnibus (GEO) from the National Center for Biotechnology Information (GEO accession number: GSE228649).

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
