# Peer review of "Transcriptomic Analysis Reveals the Inability of Recombinant AAV8 to Activate Human Monocyte-Derived Dendritic Cells"

_ijms, 2023, doi:10.3390/ijms241310447_

Round 1

Reviewer 1 Report

In this study Masri et al. investigates the immunogenicity of AAV8 on human moDCs. The uptake of AAVs by DCs plays a central role in shaping the adaptive immune response towards rAAV antigens and thus affects the outcome of gene therapy. The authors show that rAAV8 particles are efficiently internalized by moDCs, but this uptake does not induce transcriptomic changes in moDCs in contrast to an adenoviral infection, which is able to upregulate the anti-viral pathway.

Generally, the text is clearly written and the interpretation of results is proper. There are only a few grammatical errors and misspellings. However, the quality of the manuscript suits the requirements of the journal, there are some concerns needing to be addressed:

Major concerns:

1.      I have concerns regarding the word ‘activation’ used throughout the text. The goal of viral vector administration to DCs was not to induce the activation of DCs. In contrast, using recombinant AAVs the goal would be to avoid immune responses. The effect of rAAV8 administration was a question for the authors. Since the addition of rAAV8 is not a real activating stimulus, it is not proper to call control cells as non-activated. It would be more relevant and clear to use ‘non-treated’ or ‘control’ instead.

2.      Similarly, CD80 and CD86 are costimulatory molecules, the expression of which increases upon activation. The expression of many other surface molecules are increased during the activation of DCs such as CD40, MHC molecules, but any of them are called as activation markers. CD83 is usually referred to as a maturation marker, but CD80 and CD86 are costimulatory molecules. The authors should use the correct term regarding CD80 and CD86 in the manuscript.

3.      Figure 2: The data are shown from 3 human donors. At 48h and 78h only one donor is shown of the Adenovirus-incubated moDCs. The reason for that should be clarified in the text and in the figure legend.

4.      Table 2 is redundant, therefore it should be moved to supplementary material.

5.      Figure 4 is not clear at all. Why are the columns for the NA moDCs and rAAV8-infected moDCs “washed together”? Even though the results obtained from untreated cells are similar to the ones from rAAV8-infected cells each treatment should appear separately on the heat map. Please clarify.

6.      It is not clear either how many donors were pooled for 3'seq-RNA Profiling. According to the abstract the transcriptome analysis was performed on 12 healthy donors. However, it is not mentioned later on either in the main body of the manuscript or in the figure legend of Figure 4.

7.      MoDCs were stimulated with two different TLR ligands (LPS- TLR4, and R848-TLR7/8) simultaneously. The authors should explain their choice of stimulation in the manuscript. Would not be R848 alone not enough to mimic viral infection? Why was the addition of LPS, which is a bacterial cell wall component necessary? The reason behind the choice of TLR stimulation should be explained in the manuscript.

8.      Supplementary figure 1. The description of ‘a’ and ‘b’ should be clearly stated in the figure legend.

9.      lane 78: The agonists of TLR10 are largely unknown. Some studies suggest that TLR10 may heterodimerize with TLR2 and might bind the same ligands as TLR2. TLR10 may also sense RNA-protein complexes or dsRNA (doi:10.1111/sji.12988). Due to the uncertainty of TLR10 ligand preferences, I would not mention it among those TLRs that do not detect nucleic acids.

Minor suggestions:

1.      I suggest the use of 106 instead of 1e6

2.      The abbreviation of some expressions are missing such as cGAS, IF16, AIM2, RIG-1 in lane 77 or MDA5 in lane 90.

3.      The abbreviations do not always appear where they are mentioned first in the text. I.e. TLR9 appears in lane 82, though TLRs are mentioned earlier in lane 77.

4.      Figure legend 1: The meaning of NA is not stated. The meaning of ** appears in the legend, however there is no data in the bar graphs with two asterisks.  

5.      Lane 113: instead of “in presence of IL-4” use “in the presence of IL-4”

6.      Lane 423: The source of the TLR ligands LPS and R848 should be mentioned in the Material and Methods.

7.      Lane 462: the use of ‘systematically’ is redundant

8.      MoDC is the shortening of monocyte-derived dendritic cells, thus the term “moDC cell” is not proper since “cell” is already present in the abbreviation.

The quality of English is good, only a few grammatical mistakes and misspellings need to be corrected. 

Reviewer 2 Report

The authors performed transcriptomic experiments to investigate the activation of human monocyte-derived dendritic cells by AAV8. They found that AAV8 particles did not induce any detectable transcriptomic changes in moDC cells, although AAV8 particles were efficiently internalized into the moDC cells within 24 hours. These results are important for gene therapy to understand the immunogenicity of recombinant AAV vectors. The citations are well chosen. In principle, this study will be useful for readers of IJMS, but I have some additional comments below.

Major comments:

1. Lack of direct comparison with other serotypes

While the authors provide important results suggesting a low activation of innate immunity by AAV8, the lack of direct comparison with other serotypes greatly weakens their value. It is a great pity that the authors did not include other serotypes of AAV such as AAV2 or AAV9, in Figures 3 and 4.

2. Tissue tropism of AAV8

AAV8 has been shown to effectively transduce and deliver genes to the liver (Kattenhorn, 2016-Human Gene Therapy). The liver is an immunologically active organ, with large populations of phagocytic cells that play a critical role in immune activation. Viral targeting to these tissues can induce immune responses such as liver toxicity, reducing the safety and limiting the efficacy of systemic injection (Goertsen, 2022-Nat Neurosci). Therefore, even if AAV8 has a low immunogenic effect on moDC, it may not be sufficient to reduce the overall immunoreactivity caused by AAV delivery.

Minor comments:

p3, line 132 - 134

Cells were harvested at 48h post infection and reporter transgene (GFP) expression was examined by fluorescence microscopy.

I am not sure if 48 hours is long enough for AAV expression. It is well known that AAV vectors require a long time for expression (more than 1 week) because annealing process of single-strand DNA is required for gene expression.

p5, line 193 (Figure 2)

Δvg/dg (+PK)-(+PK)

Does it mean “Δvg/dg (+PK)-(-PK)”?

Please describe “dg” briefly here.

p6, line 239

DCs collected fro[42]m

Modify the reference insertion site.

p6, line 246

after 4 hours of incubation (Oshima 2009, Mol Ther)

Is this reference [28]?

p7, line 251 (Figure 3)

ssc

Does ssc used in Figure 3 mean “side scatter”? Please describe it briefly in the text.

p9, line 314 - 316

Color scheme is based on z-score distribution, from −4 (Blue) to 4 (Red) expression distributions of genes across moDC conditions on the y-axis.

Please add color scale bar in Figure 3b to clearly show upregulation and downregulation.

p10, line 350

rAAV8 can induce a humoral a to a lesser extent a cellular immunity

Typo?
